# Reasoning over Bayesian Networks using Semantic Artificial Neural Networks

Sotirios Batsakis[1,2(✉)] and Grigoris Antoniou[1]

[1] University of Huddersfield, UK
[2] Technical University of Crete, Greece
{s.batsakis,g.antoniou}@hud.ac.uk

**Abstract.** Representation of application domains, related concepts and their dependencies is often achieved using Bayesian Networks. In Bayesian Networks nodes represent random variables and arcs represent their dependencies. Since inference over Bayesian Networks is a complex task in this work a novel approach for representing and reasoning over Bayesian Networks using Semantically labeled Neural Networks is proposed and evaluated. Using Semantic Neural Networks combines advantages of Neural Networks such as wide adoption and highly optimized implementations while preserving the interpretability of Bayesian Networks which is an important requirement, especially in medical applications.

## 1 Introduction

Bayesian Networks are widely used for representation and inference tasks in numerous application areas for evaluating joint probabilities of random variables given values of evidence variables[4]. Inference over Bayesian Networks is achieved using either exact inference methods (which can be computationally expensive) or approximation inference methods. The Bayes Network parameters are usually estimated using data and the structure of the network is typically defined using domain expert knowledge, so this is a typical knowledge engineering task.

Probabilities of random variables given evidence data can be also estimated using neural networks. Neural Networks in recent years have widespread adoption in practice achieving very high performance in various tasks, but they are not an interpetable approach as required in application domains such as medical diagnosis. A solution to this problem is defining the structure of a neural network using a domain theory in the form of logic rules for creating Neural Networks with semantically labeled hidden layer nodes called Knowledge Based Artificial Neural Networks (KBANN) [7]. A variant of KBANNs, called Semantic Artificial Neural Networks (SANN) where the network structure and semantic labelling of nodes is defined using a domain representation in a graph form (e.g. a knowledge graph) has been proposed in [1].

In this work a representation of Bayesian Networks and the corresponding reasoning mechanism using Semantic Neural Networks is proposed, based on the observation that Bayesian Networks have a graph form that can be used for defining Semantic Neural Networks. Specifically a mapping algorithm from Bayesian Networks to Semantic Artificial Neural Networks is proposed and the performance of resulting neural networks is

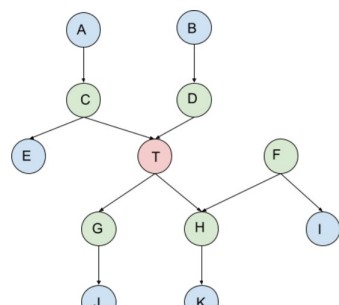

**Fig. 1.** Bayesian Network example, the Markov Blanket of node T (red) is the set of green nodes.

compared both with typical dense Neural Networks and Bayesian networks. By using such a representation the semantic structure of Bayesian Networks is preserved, thus retaining interpetability.

## 2    Representation of Bayesian Networks using Semantic Artificial Neural Networks

Bayesian Networks represent random variables in a form of Directed Acyclic Graphs were arcs between nodes represent dependencies between the corresponding random variables. Typically the arcs represent causality relations and an example Bayesian Network is presented in Figure 1. In Figure 1 arrows represent causality relations (e.g. D influences T) and T is the target feature to predict. In a Beyesian network a node is influenced by its parents, children and children's parents directly (this set of nodes is the Markov blanket of a node and the node is independent of other nodes given its Markov blanket). The Markov blanket of node T is the set C,D,F,G,H and if these nodes are known the rest can be ignored.

A Bayesian Network (BN) can be represented using a Semantic Neural Network by defining a mapping that preserves the semantic labeling of the Bayesian Network nodes. Using a Neural Network may have advantages over Bayesian Networks in terms of performance. In addition the learning process when parameters are learned from raw data is straightforward in case of Neural Networks. Furthermore if the Neural Network is a Semantic Neural Network, interpetability is preserved. The mapping algorithm is defined as follows:

An example application of the mapping algorithm is the following: Given the Bayesian Network of Figure 1 the variable to predict is $T$. Also the dataset consists of values for features T, C,H, F, B, A and J. Given the Bayesian Network and the dataset the mapping algorithm produces as output the Semantic Neural Network of Figure 2. After constructing the SANN, training the neural network can be used for creating a prediction model for the values of the target values given the remaining relevant features into the dataset. Also notice that if the target feature in the dataset changes then another SANN must be constructed, thus the BN and the corresponding dataset is represented using a set of SANNs one for each output feature.

---

**Algorithm 1** Semantic Artificial Neural Networks Construction using a Bayesian Network

---

**Require:** Dataset $D$,
**Require:** Bayesian Network (Domain Conceptualization) $BN$
 1: Create empty Neural Network Graph $G$
 2: Add node $t \in BN$ to predict as output node in $G$
 3: **for all** Input features $i_j \in BN$ that are into the Markov Blanket of node $t$ **do**
 4:     **if** $i_j \in D$ **then**
 5:         **if** $i_j \notin G$ **then**
 6:             Add $i_j$ in $G$
 7:         **end if**
 8:         Add link from $i_j$ to $t$ in $G$
 9:     **else**
10:         Apply recursively the algorithm with $i_j$ as the node to predict and expand $G$ with the sub-tree rooted in $i_j$
11:     **end if**
12: **end for**
13: **return** Graph $G$

---

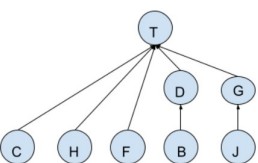

**Fig. 2.** Semantic Artificial Neural Network example

## 3   Evaluation

In this section a set of Bayesian Networks representing medical domains and the corresponding datasets are represented using SANNs following the procedure defined in Section 2. Then the performance of the resulting SANNs is compared with the performance of Bayes Net and (dense) Neural Networks for the same datasets. The datasets[3] are synthetic medical datasets with 10K samples each, for which corresponding Bayesian Networks are available. Specifically the datasets are represented using the Bayesian Networks:ALARM [2] designed to provide alarms during patient monitoring in anesthesia, ASIA [5] that models lung cancer in patients from Asia and CHILD [6] for diagnosing congenital heart disease in babies.

For all three datasets the corresponding Semantic Neural Network is constructed given a feature to predict and the Bayesian Network for this dataset following the procedure defined in Section 2. Then the performance of the resulting Semantic Artificial Neural Network (SANN) is compared with that of Bayesian Network (BN) and typical Neural Network (NN) using the corresponding implementations of WEKA machine learning software  [3].

---

[3] Available at: https://www.ccd.pitt.edu/wiki/index.php/Data_Repository

**Table 1.** Comparison (classification accuracy comparison between SANNs, NNs and BNs for the ALARM,ASIA and CHILD datasets

| Dataset/Metric | Bayesian Network | Neural Network | Semantic Neural Network |
|---|---|---|---|
| ALARM | 73.24 | 78.01 | **80.04** |
| ASIA | **86.14** | 86.11 | **86.14** |
| CHILD | 69.72 | 67.77 | **71.23** |

The experimental results presented in Table 1 (best results are highlighted in bold) indicated the high performance of SANNs compared with Bayesian Nets and dense Neural Networks. The positive results indicate the potential applicability of Semantic Neural Networks since they combine the semantic structure and explainability of Bayesian Networks while achieving the high performance and scalability of Neural Networks.

## 4    Conclusion

In this work a method for representing and reasoning over a Bayesian Network combined with a corresponding dataset using Semantic Artificial Neural Networks is proposed. Using SANNs offers an alternative solution for both problems of parameter estimation and inference of Bayesian Networks while retaining the explainability of the initial Bayesian representation. Experimental results indicated the high performance of the proposed approach. Future work can include wide scale applications of the proposed approach over large scale and complex domains especially medical domains where explainability is a strict requirement.

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
