# OpenReview forum: "Reasoning over Bayesian Networks using Semantic Artificial Neural Networks"
_eswc-conferences.org/ESWC/2021/Conference/Poster_and_Demo_Track — Submitted to ESWC2021 P&D_

### Official Review · AnonReviewer2 · 2021-04-12
**Potentially interesting contribution not fitting the scope of the conference**

**Rating:** 2
**Confidence:** 2

**Review:**

The paper introduces a method for reducing Bayesian Networks (BN) to Semantic Artificial Neural Networks (SANN). This method is the main contribution of the paper. Authors evaluate the results of classification using 3 classifiers (all implemented by authors): BN, SANN, and "typical Neural Network".

Unfortunately, I struggle to see the relation of this paper to the topic of the conference. The method does not involve any semantics or semantic web, the output is a neural network. Because the original BN is interpretable, the output NN is also interpretable, however, this feature is not exploited in the paper.

Also, though the resulting (SA)NN gives better results on some datasets, it is not actually clear and not investigated further why this happens. One could imagine the more developed optimization algorithms might be responsible for this improvement. Since the datasets are openly available, it would be useful to compare the results in this paper with the results of some other authors.

**Anonymity:**

Yes, I would like my review to remain anonymous.

---

### Official Review · AnonReviewer4 · 2021-04-14
**self-contained paper**

**Rating:** 7
**Confidence:** 1

**Review:**

This paper further elaborates on the idea of Semantic Artificial Neural Networks, which is conceptually very needed line of research.
The paper, although necessarily brief, contains all elements of a research paper. The authors have even managed to present pseudocode, example and a mini-benchmark on three datasets.

Points for improvement:
Many important details are missing, such as the setting of NN for the experiments. NN could have obtained better results if metaparameter tuning was performed.  While this won't fit into the page limit, the authors could consider adding a link to a public repository to code for replication.
The authors should clarify delta against [1].
No discussion of limitations, no evaluation of explainability.
I am not sure the presented approach should be called "reasoning".

Minor notes: The paper contains several typos (e.g., Beyesian).


**Anonymity:**

Yes, I would like my review to remain anonymous.

---

### Official Review · AnonReviewer1 · 2021-04-15
**Descriptions are vague**

**Rating:** 5
**Confidence:** 2

**Review:**

This paper proposes an approach for representing and reasoning over Bayesian Networks using Semantically
labeled Neural Networks is proposed and evaluated. It is an interesting problem however I have few concerns.

- Very unclear description of the problem.
- How Figure 2 was obtained from Figure 1.
- The paper has an Algorithm 1 but it was never referred to or explained line by line.
- Many things in the paper are very vaguely described.

**Anonymity:**

Yes, I would like my review to remain anonymous.

---

### Decision · Program_Chairs · 2021-04-19

Reject